# Furosine Posed Toxic Effects on Primary Sertoli Cells through Regulating Cep55/NF-κB/PI3K/Akt/FOX01/TNF-α Pathway

**DOI:** 10.3390/ijms20153716

**Published:** 2019-07-30

**Authors:** Huiying Li, Bingyuan Wang, Huaigu Yang, Yizhen Wang, Lei Xing, Wei Chen, Jiaqi Wang, Nan Zheng

**Affiliations:** 1State Key Laboratory of Animal Nutrition, Institute of Animal Science, Chinese Academy of Agricultural Sciences, Beijing 100193, China; 2Key Laboratory of Quality & Safety Control for Milk and Dairy Products of Ministry of Agriculture and Rural Affairs, Institute of Animal Science, Chinese Academy of Agricultural Sciences, Beijing 100193, China; 3Laboratory of Quality and Safety Risk Assessment for Dairy Products of Ministry of Agriculture and Rural Affairs, Institute of Animal Sciences, Chinese Academy of Agricultural Sciences, Beijing 100193, China; 4Shanghai Applied Protein Technology Co., Ltd., Shanghai 200030, China

**Keywords:** furosine, lipid metabonomics, PE(18:0/16:1), lactoferrin, sertoli cells

## Abstract

As one of the Maillard reaction products, furosine has been widely reported in a variety of heat-processed foods, while the toxicity of furosine on the reproductive system and related mechanisms are unclear. Here, we constructed an intragastric gavage male mice model (42-day administration, 0.1/0.25/0.5 g furosine/Kg body weight per day) to investigate its effects on mice testicle index, hormones in serum, and mice sperm quality. Besides, the lipid metabonomics analysis was performed to screen out the special metabolites and relatively altered pathways in mice testicle tissue. Mice primary sertoli cells were separated from male mice testicle to validate the role of special metabolites in regulating pathways. We found that furosine affected testicle index, hormones expression level and sperm quality, as well as caused pathological damages in testicle tissue. Phosphatidylethanolamine (PE) (18:0/16:1) was upregulated by furosine both in mice testicle tissue and in primary sertoli cells, meanwhile, PE(18:0/16:1) was proved to activate Cep55/NF-κB/PI3K/Akt/FOX01/TNF-α pathway, and as a functional protein in dairy products, lactoferrin could inhibit expression of this pathway when combined with furosine. In conclusion, for the first time we validated that furosine posed toxic effects on mice sperms and testicle tissue through upregulating PE(18:0/16:1) and activating Cep55/NF-κB/PI3K/Akt/FOX01/TNF-α pathway.

## 1. Introduction

As a classical reaction in food heat processing, Maillard reaction generates multiple of side-products [1,2], including fructolysine (FL), *N*(ε)-2-furoylmethyl-l-lysine (furosine), pronyl-lysine, pyrraline, *N*(ε)-carboxymethyl-l-lysine (CML), pentosidine, *N*(6)-(1-carboxyethyl)-l-lysine (CEL) and hydroxymethylfurfural (HMF), etc. [3,4] Being generated in the early-middle stage of Maillard reaction, furosine (C_12_H_18_N_2_O_4_, Mw 254.28) has been detected in a lot of food items, including infant formulas, dairy products, cereals, honey, coffee, tomato products, and bakery products. [5,6]. Higher furosine concentrations have been reported in infant formulas which ranged from 471.9 mg/100 g to 639.5 mg/100 g [7,8,9]. Particularly, considering furosine is a by-product in heat treatment of food, it is widely used as a marker of protein loss and food nutritional quality [10].

Though the quantitative detection technologies of furosine in foods are widely utilized and improved upon, investigation of its biological effects and metabolic courses have not been well established. In our recent study, furosine was proved to pose toxic effects on mice liver and kidney through causing downstream inflammation. However, there seems no published article elucidating the toxic effect of furosine in reproductive models, its toxicity on animal senital system and later generations is still unclear, not to mention the related mechanisms. In the present paper, the male mice were orally administrated with furosine for 42 days to construct in vivo model. To investigate the related mechanisms, lipid metabonomics detection of mice testicle tissue was performed and the special metabolites were screened out, and the effect of the metabolite sponsor on factors of Cep55, NF-κB, PI3K/Akt, FOX01, and TNF-α was investigated both in separated primary sertoli cells (SCs) and testicle tissue. To evaluate the overall toxicity of furosine on production and maturation of sperms, testicle index calculation, pathological staining of testicle tissue, reproductive hormones detection, and sperm quality test were also performed. Considering furosine widely exists in food products, it is vital to elucidate its toxicity on the genital system of mammals, which will provide theoretical basis for the limit of daily intake, as well as provide practical instructions for the thermal processing regulation. 

In this study, we evaluated the effect of oral administration of furosine on the lipid metabonomics of testis as well as on the sertoli cells. We also determined the effect of oral administration of furosine on the sperm parameters of male mice. The results showed that the special metabolite PE(18:0/16:1) was significantly up-regulated by furosine, which might in turn lead to the testis inflammation and reproductive toxicity via Cep55/NF-κB/PI3K/Akt/FOX01/TNF-α signaling pathway. Moreover, as a functional protein in dairy products, the effect of lactoferrin in regulating Cep55/NF-κB/PI3K/Akt/FOX01/TNF-α pathway was also detected and evaluated, and the results proved that lactoferrin might alleviate the adverse effect of furosine through inhibiting this pathway when combined with furosine, which deserved further experimental verifications on our part. 

## 2. Results

### 2.1. Furosine Affected Testicle Index of Male Mice. 

With 42-day administration of furosine, the testicle index of male mice increased in different degrees, and the one in 0.5 g/Kg group showed statistical significance with the control (*p* < 0.05) (Figure 1A,B), suggesting that furosine might pose toxic effects on male mice reproductive system. 

### 2.2. Furosine Affected Pathological Condition of Testicle Tissue and Reproductive Hormones Level in Serum

Utilizing HE staining, the pathological changes of testicle tissue were observed as follows. With treatment of furosine (0.25 g/Kg and 0.5 g/Kg), cytomorphosis, hyaline degeneration, inflammatory cell infiltration, disorder-arrange of seminiferous tubules, and even occasional hemorrhage could be found in testicle, especially in 0.5 g/Kg group (Figure 2A). Meanwhile, reproductive hormones in male mice serum including testosterone (T), follicle stimulating hormone (FSH), and luteinizing hormone (LH) were detected. As Figure 2B showed, these three hormones were upregulated significantly in furosine-treatment groups in a dose-dependent manner (*p* < 0.05). 

### 2.3. Special Metabolites Screening and Data Analysis

Through lipid metabonomics detection and data analysis, 37 up-regulated metabolites were screened out, mainly including diacylglycerol (DG), triglyceride (TG), phosphatidylethanolamine (PE), phosphatidylcholines (PC), meanwhile, 9 down-regulated metabolites were selected, which all with Variable Importance in Projection (VIP) > 1 and P value < 0.05 (Appendix A). Through literature searching, PE (18:0/16:1) was further determined as the special metabolic sponsor in testicle tissue, for its possible toxic effects and regulation of the related pathway (Figure 3).

### 2.4. Seperation and Culturing of Sertoli Cells 

As Figure 4 shows, sertoli cells separated from mice testes tissue were cultured in 6-well plates, and these cells demonstrated slimline type and attachment-inhibited growth, which promised the successful construction of the primary sertoli cell model. By staining with 4′,6-diamidino-2-phenylindole (DAPI), the primary sertoli cells were found to be attached on the culturing wells. By staining with Wilms Tumor 1 (WT1, a special marker of sertoli cell), the cells were proved to be alive and in normal growth condition. By staining with DAPI and Wilms Tumor 1, alive sertoli cells were found to be with high purity and viability. 

### 2.5. Expression of Cep55/NF-κB/PI3K/Akt/FOX01/TNF-α in Testicle Tissue 

Referring to the potential role of Cep55 and its downstream factors in regulating spermatogenesis and sperm quality, the expression of these factors in testicle tissue was measured by q-PCR and western blotting. The mRNAs level of Cep55, NF-κB, PI3K, Akt, FOX01, and TNF-α were up-regulated significantly in a dose-dependent manner (*p* < 0.05) (Figure 5A). The protein level Cep55, NF-κB, p-PI3K, p-Akt, p-FOX01, and TNF-α increased significantly (*p* < 0.05), while PI3K and Akt proteins seemed no obvious change, indicating phosphorylations of PI3K/Akt and FOX01 were essential for activation of these proteins (Figure 5B,C). The results suggested furosine could activate Cep55/NF-κB/PI3K/Akt/TNF-α/FOX01 pathway, and might pose toxic effects through inducing downstream inflammations. 

### 2.6. Expression of Cep55/NF-κB/PI3K/Akt/FOX01/TNF-α Affected by Furosine and Special Metabolite in Sertoli Cells

Utilizing the normal primary sertoli cells separated from control mice to be treated with furosine or PE(18:0/16:1) or lactoferrin for 48 h, Cep55/NF-κB/p-PI3K/p-Akt/Fox01/TNF-α proteins were detected by western blotting. As Figure 6A,C showed, compared to the control, Cep55, p-FOX01, NF-κB, p-PI3K, p-Akt and TNF-α proteins were markedly up-regulated both in furosine treatment group and in PE(18:0/16:1) treatment group (*p* < 0.05); meanwhile, there seemed no difference of the increase degree between the above two groups, indicating that PE could activate the pathway and furosine affected spermatogenesis and sperm quality probably through regulating the level of PE and the related pathway. Moreover, single lactoferrin treatment did not affect the levels of these proteins; however, the pathway was significantly inhibited in the combination groups (furosine+lactoferrin or PE(18:0/16:1)+lactoferrin), when compared with the ones in furosine group or PE(18:0/16:1) group (*p* < 0.05) (Figure 6A,C).

Utilizing the sertoli cells separated from mice treated with furosine for 42 days, Cep55, p-FOX01, NF-κB, p-PI3K, p-Akt, and TNF-α proteins proved to increase in treatment groups (Treatment (T), T+Furosine, T+PE (18:0/16:1)) comparing to the control (*p* < 0.05) (Figure 6B,D), further validated that furosine affected genesis and quality of sperm through regulating PE in sertoli cells and activating Cep55/NF-κB/PI3K/Akt/FOX01/TNF-α pathway.

### 2.7. Sperm Quality Affected by 42-day Treatment of Furosine 

Furosine administration affected several parameters of male mice sperms (Table 1). Compared to the control, the percentage of progressive sperms showed no obvious difference in furosine treatment group. VAP (average path velocity), VSL (straight line velocity) and VCL (curvilinear velocity) of sperms significantly decreased with treatment of furosine for 42 days when comparing with the control (*p* < 0.05). The above results suggested that furosine posed adverse effects on sperm quality, which might result in poor development of mouse descendants.

All the data were represented as mean ± SD, *n* = 8. * Comparing with the control, *p* < 0.05.

## 3. Discussion

Though furosine is a widely existed by-product in multiple foods and the detection technologies have been improved for several times, the toxicity of furosine was rarely revealed, especially the toxic effect on the reproductive system [11,12,13]. Sperm genesis is a primary status of the formation of embryos, which are essential for the quality and development of embryos, so clinical sperm quality detection becomes a common technique in reproductive medicine area to diagnose the proper period for breeding [14,15]. Considering furosine is a conventional chemical in milk, formulas, baked desserts, honey, coffee, etc., excess intake of these foods might be harmful for the males, even affect the health and development of the younger generation. 

In the present study, male mice model administrated with furosine was constructed and its toxicity on reproductive system was proven. The sperm quality was measured, then lipid metabonomics detection was exerted to screen out the special metabolites in testicle tissue. To further reveal the role of furosine in regulating Cep55/NF-κB/PI3K/Akt/FOX01/TNF-α pathway, the primary sertoli cells were separated and cultured, and these factors were detected in sertoli cells both in mRNA level and in protein level.

Centro-somal protein 55 (Cep55) was found in 2004 by Fabbro, which belongs to the centrin family and expresses in the testes, spleen, liver, kidney, and colon, and its peak expression level was found in testes and thymus tissue [16,17]. Cep55 was proved to play a key role in regulating cell memebrane division and confusion through combining with testes expressed 14 protein (TEX14) before meiosis [18]. Research found that overexpression of Cep55 caused severe progressive defects in growth course of male mice sperm, which might be related to sperm growth stagnation and sperm cell lackage in testes tissue [19].

Forkhead protein in rhabdomyosarcoma (FOX01) widely exists in the liver, brain, lung, kidney, liver, testes, and intestine in mammals, which expresses differently at various stages of growth and development and affected by growth factors, cytokines and oxidative stress [20]. Research found that FOX01 was activated by ROS and then regulated by phosphatidylinositol 3-kinase/protein kinase B (P13K-PKB/Akt) [21,22,23]. Sedding also found that H_2_O_2_ upregulated phosphoralation of FOX01 by activating Akt, which might be achieved by decreasing several antioxidative genes, like MnSOD and CAT [24,25].

D. Sint et al. also found that overexpression of Cep55 resulted in activation of PT3K/Akt pathway and increased phosphoralation of FOX01 protein in testes, which led to sperm cell inactivation and degenerative reproductive capacities [19]. Therefore, in the present study, factors of Cep55, NF-κB, PI3K/Akt, FOX01, and TNF-α were detected in testes and primary sertoli cells at both mRNA level and protein level, aiming to investigate the role of furosine in regulating this pathway and affecting sperm quality through special metabolite PE(18:0/16:1). There seemed no difference in the activating degree of these factors in furosine treatment group, PE(18:0/16:1) treatment group and furosine+PE(18:0/16:1) group, verified that furosine posed toxic effects on sperms by activating Cep55/NF-κB/PI3K/Akt/FOX01/TNF-α pathway through PE(18:0/16:1).

Considering the spermatogenesis course depends on the combination of T and androgen receptor (AR) on genital cells [26], LH can combine with luteinizing hormone receptor (LHR) on interstitial glands (Leydig cells) [27], FSH can take effect on sustentacular cells of testis (sertoli cells) and spermatogenic cells through binding with follicle stimulating hormone receptor (FSHR) of sertoli cells [28], all the three types of receptors play roles in regulating harmones in testicle, furosine might affect production and quality of sperm through regulating reproductive harmones.

The mammalian testis consists of seminiferous tubules and surrounding interstitial tissues, which includes Leydig cells, blood vessels, leukocytes, and fibroblasts. Seminiferous tubules surrounded by peritubular myoid cells are the sites where spermatogenesis occurs and are composed of sertoli cells (SCs) as well as different stages of male germ cells [29]. The constant spermatogenesis relies on the spermatogonial stem cells (SSCs) which keep self-renewing to maintain the pool of stem cells and then differentiate into several types of spermatocytes and eventually spermatozoa [30,31]. Both SSCs and SCs locate on the basement membrane of seminiferous tubules. As a key component of SSCs microenvironment, SCs play crucial roles in the regulation of the self-renewal and differentiation of SSCs by secreting a variety of growth factors and cytokines [32,33,34,35,36,37,38]. Moreover, SCs protect male germ cells by forming SC barrier/blood–testis barrier (BTB), thereby establishing an immune privileged microenvironment [39]. SC population reduction or maturation arrest is closely related to abnormal spermatogenesis [40,41]. Further, failure of connection between SCs as well as between SCs and germ cells also leads to spermatogenic defects and infertility in transgenic mice [42,43]. Therefore, SCs serve as nurse cells to provide structural, immunological, and nutritional support for SSCs, which is critical for spermatogenesis and reproductive functions [44]. 

Seminiferous tubules contain genital cells and sertoli cells, which exist in inner wall of the tubules. Another type of cell named mesenchymal cells (leydig cells) locate at the outside of seminiferous tubules, in around connective tissues. The above three types of cells support the normal function of testicle [45,46,47]. In the present manuscript, pathological results of testes tissue, metamorphose, and edema of seminiferous tubules, as well as infiltration of inflammatory cells around these tubules, combining with decreased concentration of progressive sperms and exasperated sperm quality, demonstrated that furosine caused the damages of seminiferous tubule and functional cells, then took adverse effect on sperm quality finally.

To investigate the special metabolic sponsor in testicle tissue and the specific upstream target of Cep55-related pathway, lipid metabonomics detection was exerted and the data showed that PE(18:0/16:1) was proven to be the specific metabolite in testicle tissue, which could activated expressions of Cep55, NF-κB, PI3K/Akt, FOX01, and TNF-α both in testes tissue and in primary sertoli cells. Considering PE(18:0/16:1) is produced from the reaction of ethanolamine and 1, 2-diglyceride in animals, and can be transferred to lysophosphatidyl ethanolamine 18:0 (LPE(18:0)) by phospholipase A [48]. Research found that PE family was the essential component in the production course of mammal sperm [49], the drastic loss of PE family and several polyunsaturated fatty acids in sperm could be an important cause of male infertility, as infertility of men with normal semen quality could originate from the disorder of sperm lipid metabolism [50]. However, the research of the effect of PE(18:0/16:1) on sperm quantity and quality was rare, the transfer reaction of PE(18:0/16:1) and LPE(18:0) in animals was also unclear, our study for the first time reported that PE(18:0/16:1) could be upregulated by furosine in testicle and could activated Cep55-related factors, which finally took adverse effect on sperm formation and development. Nevertheless, the toxicokinetic research of PE(18:0/16:1) in furosine-administration model in vivo requires further attention.

As an iron-binding protein with approximately 700 amino-acid residues, lactoferrin (LF, 80 kDa) was widely found in milk, tears, seminal fluid, and saliva which are usually secreted by mammals [51,52]. Due to the two Fe^3+^ binding sites in its structure, lactoferrin is divided into three types, the apo-type (without iron atom), the single iron type (binding with 1 iron atom), and the holo-type (binding with 2 iron atoms). Lactoferrin was proven to play key roles in alleviating inflammation, oxidation damages, viral infections, and various types of tumors [53,54,55,56,57,58,59,60]. Therefore, the addition of lactoferrin in foods, including milk powder, functional beverage, and some health products became increasingly widespread. However, research of the protective effect of lactoferrin in reproductive system was rarely seen. Further to validate the role of lactoferrin in regulating Cep55/NF-κB/PI3K/Akt/FOX01/TNF-α pathway, as well as to investigate the possible role in alleviating the adverse effect of furosine, we detected the expression of this pathway in sertoli cells by western blotting. Results demonstrated that furosine+lactoferrin combination or PE(18:0/16:1)+lactoferrin combination significantly inhibited the levels of Cep55, NF-κB, p-PI3K, p-Akt, p-FOX01, and TNF-α, when compared with the furosine group or PE(18:0/16:1) group, while the single lactoferrin seemed no obvious effects on these factors, indicating lactoferrin might alleviate the adverse effects of furosine through regulating Cep55/NF-κB/PI3K/Akt/FOX01/TNF-α pathway in sertoli cells, when combined with furosine. The above data further suggested that the functional proteins including lactoferrin might neutralize the harmful bioactivities of furosine in dairy products, which might be a helpful explanation for that ultra-high temperature (UHT) milk products were still be overall beneficial to human health.

To conclude, the present study not only provides valid experimental data for safety assessment of furosine, but also attracts human’s more attention about the negative effects of furosine in male reproductive system, which would fasten the limit standard formulation of furosine in food, especially in dairy products.

## 4. Materials and Methods

### 4.1. Chemicals

Furosine was purchased from PolyPeptide (Strasbourg, France); the purity was above 95%. Dulbecco’s Modified Eagle Medium (DMEM), fetal bovine serum (FBS), and non-essential amino acids (NEAA) were obtained from GIBCO (Waltham, MA, USA), 1% penicillin/streptomycin was purchased from Thermo Fisher (Waltham, MA, USA). Lactoferrin was purchased from Sigma (St. Louis, MO, USA), with the purity of above 95%.

Cell counting kit-8 (CCK-8 kit) was purchased from Solarbio (Beijing, China), Elisa detection kits of T, FSH and LH in mice serum were purchased from Jiancheng (Nanjing, China). Hematoxylin-Eosin (HE) staining kit was purchased from Solarbio (Beijing, China).

Trizol buffer, RIPA lysate buffer (including proteases and PMSF), and protein concentration detection kit were obtained from Beyotime (Nanjing, China). Primers of GAPDH, Cep55, NF-κB, PI3K, Akt, FOX01 and TNF-α were synthesized from Sangon (Shanghai, China), reagents related with quantitative real time polymerase chain reaction (q-PCR) were purchased from Sangon. Primary antibodies of β-actin, Cep55, NF-κB, PI3K, p-PI3K, Akt, p-Akt, FOX01, p-FOX01, and TNF-α, as well as the secondary antibodies, were purchased from Santa Cruz Biotechnology (Santa Cruz, CA, USA). Reagents related with western blotting were purchased from Solarbio. Enhanced chemiluminescence (ECL) reagent was purchased from Tanon (Shanghai, China).

### 4.2. Animal Model Construction

CD-1 mice were purchased from Beijing Vital River Laboratory Animal Technology Co., Ltd. (Beijing, China) with the license number SCXK 2012. The animal experiments were approved by the Ethics Committee of Chinese Academy of Agriculture Sciences (Beijing, China) (Permission number: CAS20181010, 10 October 2018). 32 CD-1 male mice (20 ± 2 g) were randomly divided into four groups: control group (without any treatment), 0.1 g/Kg b.w. furosine group, 0.25 g/Kg b.w. furosine group, and 0.5 g/Kg b.w. furosine group. All the mice in treatment groups were orally administrated with furosine once per day, for consecutive 42 days. On day 43, the male mice were euthanized, testicle tissue and blood sample were gathered. Organ index calculation as: weight of organ / b.w. × 100%.

### 4.3. Lipid Metabolism Detection and Data Analysis

50 μg testicle tissue was treated with 500 μL isopropanol (IPA, Solarbio), after vortex (10 s) and sonicatig (10 min), the mixture was frozen under −20 °C for 1 h and then centrifuged (10,000 rpm, 10 min). The upper layer was collected and dried utilizing nitrogen. The dried samples were re-dissolved in initial mobile phase of UPLC and transferred into sample vials to be injected and analyzed by ultra-performance liquid chromatography quadrupole time of flight mass spectrometry (UPLC QTOF MS).

The UPLC QTOF MS analysis was performed by a UPLC system (ACQUITY UPLC I-Class, Waters, Manchester, UK), which was coupled with an electrospray ionization quadruple time-of-flight mass spectrometer (ESI-QTOF MS) (Xevo G2-S Q-TOF, Waters, Manchester, UK). A reversed phase LC column (Acquity UPLC CSH C18 Column, 130 Å, 1.7 mm, 1 mm × 100 mm, 1/pkg) was utilized for separation with two solvents: ‘A’ comprising acetonitrile and water (3:2) with 10 mM ammonium formate and 0.1% formic acid, and ‘B’ comprising isopropanol and acetonitrile (9:1) with 10 mM ammonium formate and 0.1% formic acid. The UPLC autosampler temperature was set at 10 °C and the injection volume for each sample was 2 mL. Column temperature was maintained at 55 °C.

Mass spectrometry was performed in eitherpositive (ESI+) or negative (ESI-) electrospray ionization mode. The data-independent MSE scan method of Waters QTOF mass spectrometer was applied, by which both high-accuracy primary MS and secondary MS/MS data of precursor ions can be automatically acquired. The MSE data were acquired in continuum mode from 50 m/z to 1500 m/z mass range for TOF-MS scanning. Capillary voltage of 3 kV and a sampling cone voltage of 30 V in both ionization modes were set. The desolvation gas flow was set to 500 L/h, and the temperature was set to 400 °C. The source temperature was set at 120 °C. The MSE scan was achieved in two scan functions using ramp collision energy. For low energy mode, scan range 100–1500 Da, scan time 0.2 s, and collision energy 6 V were set. While for high energy, scan range 50–1500 Da, scan time 0.2 s, and a collision energy ramp 15–45 V were employed. Accurate mass was maintained by introduction of a lock-spray interface of leucine-enkephalin (556.2771 [M + H]^+^ or 554.2615 [M − H]^−^) at a concentration of 1 pg/µL at a rate of 5 µL/min. Pooled quality control (QC) samples (generated from taking an equal aliquot of all the samples included in the experiment) were run at the beginning of the sample queue for column conditioning and every ten injections thereafter to correct retention time drifts and variation in ion intensity over time. The sample queue was randomized to remove bias. Batch acquisition was repeated to check experimental reproducibility.

High-accuracy MS data were recorded by MassLynx 4.1 software (Waters, Manchester, UK). Raw data were imported to commercial software Progenesis QI (Version 2.3, hereinafter referred to as QI) for data processing, which included peak annotation and normalization, as well as lipid identification through database search were performed by Progenesis QI software (Waters, Manchester, UK). All lipid species between groups were analyzed. The intensity of each lipid species was compared using non-parametric Wilcoxon rank-sum test (R package) to determine the difference of each metabolite between two groups. The relevant false discovery rates (FDR) based on the *P* values was calculated and *p* < 0.05 was considered significant difference. The differential lipid species were graphed box plot by graphpad prism 6.0. Multivariate statistical analysis was accomplished using orthogonal partial least squares-discriminant analysis (OPLS-DA) and Variable Importance in Projection (VIP) of each lipid species was obtained (EZinfo 3.0—internal software of Progenesis QI).

### 4.4. Seperation and Culturing of Sertoli Cells

Sertoli cells isolation was performed using a two-step enzymatic isolation procedure as previously described with some modifications [61]. Briefly, the testes from CD-1 male mice (both in control and furosine-treated mice in 0.5 g/kg group) were decapsulated and cut into small pieces followed by the first digestion with 2 mg/mL collagenase IV (Sigma) and 0.1 mg/mL DNase I (Sigma) in DMEM/F12 (GIBCO) at 37 °C for 30 min on a shaker. After washing three times with DPBS (Solarbio), the digested seminiferous tubules were further digested with 0.25% Trypsin-EDTA (GIBCO) at 37 °C for 8 min with gentle agitation. Then the digestion was terminated by DMEM/F12 containing 10% FBS. The digested cells were filtered through 100 and 40 μm mesh and cultured in DMEM/F12 containing 10% FBS for 24 h to allow sertoli cells attachment. The attached sertoli cells were further cultured in DMEM/F12 containing 10% FBS and incubated at 37 °C in an atmosphere of 5% CO_2_.

### 4.5. Cep55-related Pathway Detection By q-PCR and Western Blotting

The primary sertoli cells separated from the normal mice were treated with furosine (10 μM) or PE(18:0/16:1) (10 μM) or lactoferrin (final concentration 10 g/L) for 48 h according to the groups division (control, furosine group, PE(18:0/16:1) group, lactoferrin group, furosine+lactoferrin group, PE(18:0/16:1)+lactoferrin group). The primary sertoli cells (T in Figure 6) separated from the mice in furosine group were treated with furosine (10 μM) or PE(18:0/16:1) (10 μM) for 48 h according to the groups division (Control, T group, T+furosine group, T+PE(18:0/16:1) group). 

Then the cells were gathered, and the total RNA was extracted from testicle tissue using a TransZol Up Kit (Transgen, Beijing, China). The quantity and concentration of RNA were evaluated by 1.2% agarose gel electrophoresis and Nanodrop 2000 (Thermo fisher, Waltham, MA, USA). The total RNA was reversely transcribed into cDNA using a High Capacity cDNA Archive Kit, according to the manufacturer’s protocol. Primers for β-actin, Cep55, NF-κB, PI3K, Akt, FOX01 and TNF-α are listed in Table 2, and β-actin was chosen as the internal control. Quantitative real-time RT-PCR (qRT-PCR) was performed in 96-well plates in a total volume of 20 μL, containing 10 μL Universal Master Mix, 0.5 μL forward primer (10 μM), 0.5 μL reverse primer (10 μM), 1 μL template cDNA (cDNA, 10 ng/μL), and 8 μL RNAse-free water. All qRT-PCR reactions were performed at 94 °C for 30 s, followed by 40 cycles of 94 °C for 5 s, and 64 °C for 30 s, using two-step method. All q-PCR reactions were performed on the ABI 7900 HT system, and were conducted in triplicate to ensure methodological reproducibility.

20 mg testicle tissue or 10^6^ sertoli cells sample were added with 0.5 mL RIPA buffer, then homogenized by sonifier (Branson, USA) (on ice, 3 × 8 min) and centrifuged (4 °C, 10,000× *g* 5 min). The supernatant was gathered and total protein concentration was detected by BCA kit (Solarbio). After adjustment and heat treatment, the protein samples were added into the 10% SDS-polyacrylamide gel, and after electrophoresis, the proteins were transferred onto nitrocellulose filters by Trans-blot machines (Bio-Rad, Hercules, CA, USA), and the membrane was blocked with 2% BSA in TBST buffer for 1 h at room temperature (RT). Then the proteins were probed with specific antibodies for 2 h at RT, including β-actin, Cep55, FOX01, NF-κB, PI3K, p-PI3K, Akt, p-Akt, and TNF-α, the β-actin was utilized as the internal reference to assure the equal loadings. After three washings with PBST buffer (3 × 10 min), the membrane was incubated with secondary antibodies for 1.5 h at RT and then washed with PBST buffer (3 × 15 min). Finally, the membrane was detected utilizing an ECL reagent and analyzed by Image J software (Rawak Software, Inc. Germany).

### 4.6. Detection of Reproductive Hormone

The biochemical indicators including T (testosterone), FSH (follicle-stimulating hormone) and LH (luteinizing hormone) in mice serum were detected by ELISA kits according to the protocols, the final optical density (OD) value was measured using microplate reader.

### 4.7. Pathological Observation by HE Staining

Testicle tissue was cut into small pieces and fixed in 10% formalin for 48 h, then the tissue was embedded in paraffin and sliced by slicing machine (Leica, Germany). The slices were stained with hematoxylin and eosin according to the protocol of HE staining kit, and the slides were examined under light microscope (Olympus, Tokyo, Japan).

### 4.8. Assessment of Sperm Parameters

Cauda epididymides from the mice treated with furosine for 42 days were dissected in 1 mL pre-warmed Ham’s F10 buffer (Sigma, USA) and incubated at 37 °C for 15 min to allow spermatozoa to swim out. Sperm concentration, motility and progressive motility were determined using sperm analysis system (Tsinghua Tongfang, Beijing, China), and at least 200 spermatozoa from each sample were assessed.

### 4.9. Statistical Analysis

All the data were expressed as mean ± standard deviation (SD) from several independent experiments (n ≥ 3). Statistical analyses were performed by the software SPSS 13.0 (SPSS Inc, USA). An analysis of variance (ANOVA) and independent samples t-test were used to determine the differences among the treatments. P value less than 0.05 (*p* < 0.05) were considered to be statistically significance.

## Figures and Tables

**Figure 1 ijms-20-03716-f001:**
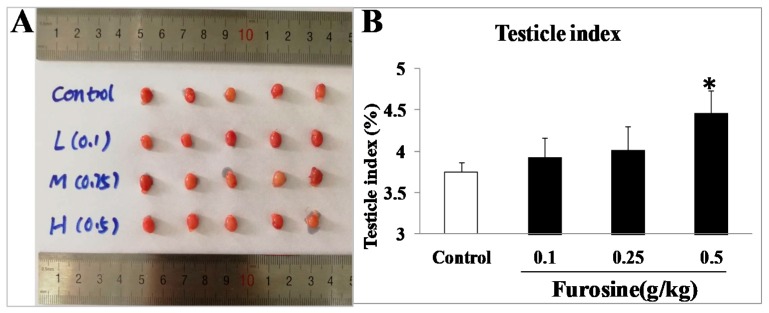
The effects of furosine on testicle index. (**A**) Mice testicles in each group. (**B**) Testicle index in each group. All the data were represented as mean ± SD, *n* = 8. * Comparing with the control, *p* < 0.05.

**Figure 2 ijms-20-03716-f002:**
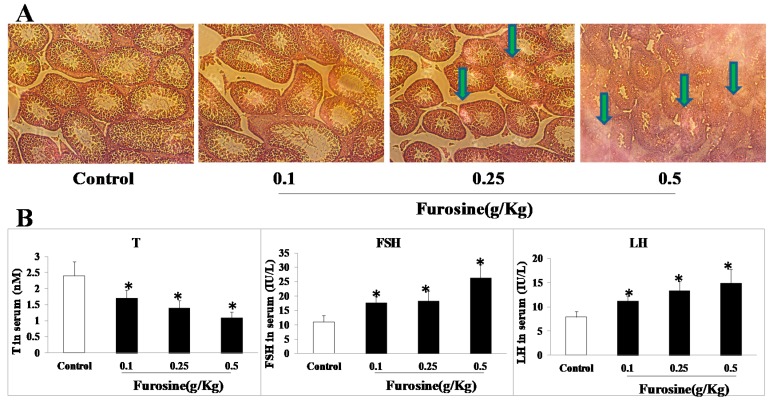
Furosine affected pathological conditions of testicle tissue and hormones in serum. (**A**) Pathological conditions of testicle tissue (×200). The green arrows stood for the apparent pathological changes of testicle tissue. (**B**) The levels of testosterone (T), follicle stimulating hormone (FSH), and luteinizing hormone (LH) in mice serum. All the data were represented as mean ± SD, *n* = 8. * Comparing with the control, *p* < 0.05.

**Figure 3 ijms-20-03716-f003:**
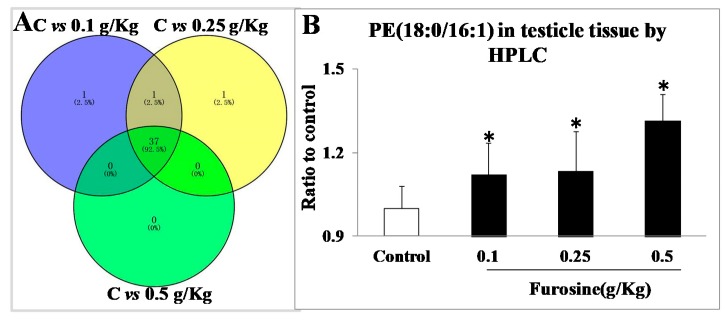
Analysis of lipid metabolites and detection of the special metabolite phosphatidylethanolamine (PE) (18:0/16:1) in testicle tissue by high performance liquid chromatography (HPLC). (**A**) VENN plot. The blue dot stands for metabolites overlapped both in control group and 0.1 g/Kg group, the yellow dot stands for metabolites overlapped both in control group and 0.25 g/Kg group, the green dot stands for metabolites overlapped both in control group and 0.5 g/Kg group. The special metabolites were screened out through overlapping the three areas. (**B**) PE (18:0/16:1) detection in testicle tissue by HPLC. All the data were represented as mean ± SD, *n* = 8. * Comparing with the control, *p* < 0.05.

**Figure 4 ijms-20-03716-f004:**
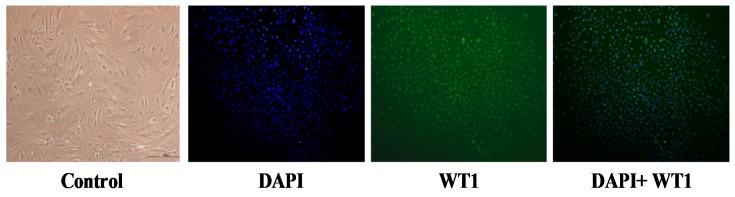
The primary sertoli cells separated from testicle tissue (Confocal). DAPI (4′,6-diamidino-2-phenylindole) stands for staining of cell nucleus, WT1 stands for staining of the special marker Wilms Tumor staining of the sertoli cells, DAPI+WT1 stands for merged registration of two stainings of the cell nucleus. The photograghs were 200 times captured.

**Figure 5 ijms-20-03716-f005:**
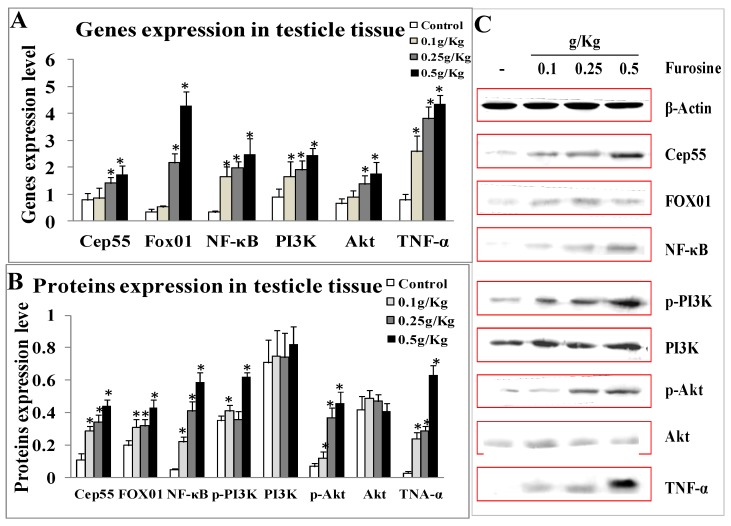
Expression of Cep55, NF-κB, (p-)PI3K/(p-)Akt, (p-)FOX01 and TNF-α at both mRNA level and protein level in testicle tissue. (**A**) Genet expressions of these factors (analyzed by software). (**B**) Proteins expressions of these factors analyzed by software. (**C**) Images of these proteins by western blotting. All the data were represented as mean ± SD, *n* = 3. * Comparing with the control, *p* < 0.05.

**Figure 6 ijms-20-03716-f006:**
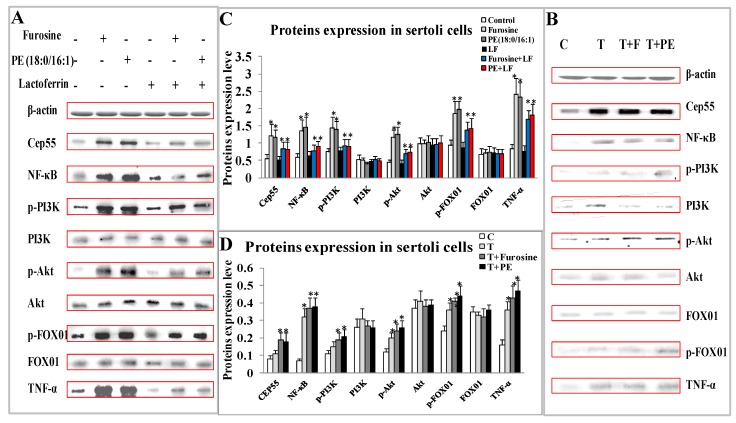
Expression of Cep55, NF-κB, (p-)PI3K/(p-)Akt, (p-)FOX01, and TNF-α proteins in primary sertoli cells. (**A**) Images of these proteins by western blotting in primary sertoli cells separated from normal mice. (**B**) Images of these proteins by western blotting in primary sertoli cells separated from furosine-treated mice, T stands for sertoli cells separated from furosine-treated mice, F stands for furosine treatment, PE stands for PE(18:0/16:1) treatment. (**C**) Protein expressions of these factors in primary sertoli cells separated from normal mice (Figure 6A, analyzed by software). (**D**) Proteins expression level of these factors in primary sertoli cells separated from furosine-treated mice (Figure 6D, analyzed by software), T stands for sertoli cells separated from furosine-treated mice, F stands for furosine treatment, and PE stands for PE(18:0/16:1) treatment. All the data were represented as mean ± SD, *n* = 3. * Comparing with the control, *p* < 0.05.

**Table 1 ijms-20-03716-t001:** Indicators of sperm quality.

Parameter	Control	Treatment
Progressive sperm (%)	52.1 ± 11.2	45.6 ± 6.9 *
VAP (μm/s)	43.8 ± 5.3	40.6 ± 9.3 *
VSL (μm/s)	28.5 ± 5.5	22.1 ± 7.6 *
VCL (μm/s)	82.7 ± 8.7	75.9 ± 18.1 *

All the data were represented as mean ± SD, *n* = 8. * Comparing with the control, *p *< 0.05.

**Table 2 ijms-20-03716-t002:** Primers of the genes in q-PCR detection.

Gene Name	Primer Sequences (5′ → 3′)	
	Forward Primer	Reverse Primer
Cep55	CGACAAGCCGTGACTCAGT	TCCTCCGACCTCTTCCTCTC
FOX01	GTTGTTGACTTCTGACTCTCCT	CCATCCTACCATAGCCATTGC
NF-κB	CAAGAGTGATGACGAGGAGAGT	GTGGAGGTGGATGATGGCTAA
PI3K	TAGGAGGAGGTTGGAAGAAGAC	CTACGGAGCAGGCATAGCA
Akt	ATGGACTCAAGAGGCAGGAAG	GCAGGACACGGTTCTCAGT
TNF-α	CGTGGAACTGGCAGAAGAG	GTAGACAGAAGAGCGTGGTG
β-actin	CCTGTATGCCTCTGGTCGTA	CGCTCGTTGCCAATAGTGAT

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
