# Peer review of "Furosine Posed Toxic Effects on Primary Sertoli Cells through Regulating Cep55/NF-κB/PI3K/Akt/FOX01/TNF-α Pathway"

_ijms, 2019, doi:10.3390/ijms20153716_

Round 1
Reviewer 1 Report
All concerns in the original review have been adequately addressed in the revised manuscript. The addition of the lactoferrin data strengthens the manuscript.
Author Response
Comments and Suggestions for Authors:
All concerns in the original review have been adequately addressed in the revised manuscript. The addition of the lactoferrin data strengthens the manuscript.
Answer: We really appreciate your help and consideration. Thank you very much!

Reviewer 2 Report
It will be helpful if the authors add the data for effect of unsaturated fatty acids on reproductive toxicity in this study.
Earlier title was better than the one in the revised version.
Author Response
Comments and Suggestions for Authors:
1 It will be helpful if the authors add the data for effect of unsaturated fatty acids on reproductive toxicity in this study.
Answer: Thank you so much for your meaningful suggestions. In our other studies, the effects of several unsaturated fatty acids like Ω-3 and Ω-6 in alleviating reproductive toxicity were evaluated and compared, which would be completed in the near future.
2 Earlier title was better than the one in the revised version.
Answer: We have replaced the present title with the earlier one, to make it more clear and comprehensive.
In conclusion, we really appreciate your suggestions and consideration. Thank you very much!

Reviewer 3 Report
Furosine does not exist in food and it is not a Maillard product, so humans are not exposed to furosine.
This research on the toxicity of furosine has no interest.
This article should not be accepted.
Author Response
Considering the toxicity research of furosine in the past three years, here, we want to add some information to explain and response to the above two questions, which is shown as follows:
“Many previous studies had shown that furosine was one of the hydrolysis products released by typical early-stage Amadori products, i.e., fructoselysine and lactuloselysine, and some of these products from severe temperature treated food had been proven to be determinated in blood [1,2], which were proved to participate the pathogenesis of diabetes and other diseases [3]. For these products always bind with proteins, herein they are proved to be undergone acid or enzymatic hydrolysis in vivo and in vitro, during this process, some Amadori products can be degraded to furosine, lysine, pyridosine, etc. [2,4,5]. Thus, furosine can be regarded as a member in Mailliard reaction products (MRPs) family as they have some similar structures, in the review of Helmut F. Erbersdobler and Veronika Somoza, furosine was also put into the list of MRPs. (please see the table in the attachment)
From this review, we also found that the content of furosine was relatively high in UHT milk and strerilized milk when compared to raw milk sample. (please see the table in the attachment)
In addition, in our experiments of renal toxicity caused by furosine and other MRPs (unpublished, under review, eLIFE), we found that furosine, pyrraline and hydroxymethylfurfural which all shared same furan ring posed toxic effects on kidney cells. Furan ring was proved to be the main toxic group to normal kidney cells via structure-activity relationship (SAR) study. Thus, like furosine, whether some other MRPs with the same groups were toxic to sertoli cells and testicle tissue through triggering the similar pathways, requiring our further study and validation, which would attract more attention in food heating industry.”
Thank you so much for your help, best regards.
References
1. RératRégine A., et al. Nutritional and metabolic consequences of the early Maillard reaction of heat treated milk in the pig. Eur. J. Nutr. 2002, 41(1): 1-11..
2. Somoza V., et al. Dose-dependent utilisation of casein-linked lysinoalanine, N(epsilon)-fructoselysine and N(epsilon)-carboxymethyllysine in rats. Mol. Nutr. Food Res. 2006, 50(9): 833-841.
3. Oimomi M., et al. Increased Fructose-Lysine of Hair Protein and Blood Glucose Control in Diabetic Patients. Hormone and metabolic research 1988, 20(10): 654-655.
4. Erbersdobler H.F., et al. Forty years of furosine–Forty years of using Maillard reaction products as indicators of the utritional quality of foods. Mol. Nutr. Food Res. 2007, 51(4): 423-430.
5. René K., et al. Studies on the formation of furosine nd pyridosine during acid hydrolysis of different Amadori products of lysine. Eur. J. Food Res. Tech. 2003, 216(4): 277-283.

This manuscript is a resubmission of an earlier submission. The following is a list of the peer review reports and author responses from that submission.
Round 1
Reviewer 1 Report
Furosine is an acid hydrolysis product formed ONLY in test tubes in laboratories. It is derived from the Amadori products present in foods.
Furosine does not exist in foods. Therefore there is no use to study the toxic effects of this molecule.
This research does not provide any benefit for the scientific community.
Reviewer 2 Report
1. Please mention full word while using abbreviations for the first time.
2. Elaborate section 2.3 and 2.4 especially role of PE in spermatogenesis.
3. Would treating the cells with unsaturated fatty acids or decreasing PE levels rescue the pathological phenotype.
4. Discuss other methods to rescue the pathological condition.
5. Please check the entire manuscript for typos and grammar.
Reviewer 3 Report
The manuscript by Li, et al entitled Furosine posed toxic effects on primary sertoli cells through regulating Cep55/NF-kB/PI3K/Akt/FOX01/TNF-alpha pathway begins to explore the effect of furosine, a frequently identified contaminant in heat-processed food, on testicular cells and function of spermatozoa. Using a chronic exposure model to furosine, the authors show that the furosine exposed mice had increased testicular size with the presence of inflammation and disarrangement of the seminiferous tubules. More importantly, exposure to furosine affected sperm motility. Using metabolic screening, it was identified that exposure to furosine increased levels of phosphatidylethanolamine (PE) in testicular tissue. Mechanistically, these effects of furosine are proposed to be regulated through the activation of the PI3 kinase/Akt pathway with increases in Cep55 and NF-kB activation and elevated levels of TNF alpha. In addition, PE is stated as having a synergistic effect with furosine. This reviewer has the following concerns.
Major:
1. In the introduction there is no discussion on the known effects of furosine on reproductive health in humans. This discussion is important in regards to the need for the present study and places it in the correct context.
2. It is unclear how exposure to furosine in male mice led to growth inhibition in fetal mice and decreased size of the placenta as the manuscript does not examine any mechanism for these effects nor discusses these results in any significant detail. Furosine exposure does affect testicular cells and sperm motility but these findings should not affect fetal growth. These experimental findings should be removed from the manuscript as they do not add to the present findings or further experiments should be performed to explain the findings on fetal development.
3. Similar to above in #2, the effects of furosine on fertility of the male mice needs to be examined and presented in the manuscript. One would expect that fertility in the furosine treated mice would be significantly impaired.
4.The metabolic profile shown if Figure 3 is only briefly described and states that 37 upregulated chemicals were identified but does provide a list of these chemicals. Such a list would be helpful to put the manuscript and its findings into context. Also, were there chemicals that were downregulated in testicular tissue after furosine exposure?
5. The findings shown in Figures 5 and 6 do suggest a role for PI3K, Cep55 and NF-kB signaling as a mechanism for their findings, however the findings are descriptive in nature. To further suggest a role of PI3K for the effect of furosine, additional experiments need to be completed wherein a PI3K inhibitor is used, or PI3K gene deficient mice, along with furosine exposure to assess if the effects are reversed.
6. The immunoblots in Figure 6 in several cases (e.g. Cep55) do not present well. These need to be improved to more clearly show the purported findings.
7. The experiments in Figure 6 that included co-exposure to furosine and PE take away from the manuscript as there is no synergistic effect upon exposure to both furosine and PE with the effect similar to the groups only exposed to one agent.
Minor:
1. There are several grammatical errors that need to be corrected.